# Spheroid-Like Cultures for Expanding Angiopoietin Receptor-1 (aka. Tie2) Positive Cells from the Human Intervertebral Disc

**DOI:** 10.3390/ijms21249423

**Published:** 2020-12-10

**Authors:** Xingshuo Zhang, Julien Guerrero, Andreas S. Croft, Christoph E. Albers, Sonja Häckel, Benjamin Gantenbein

**Affiliations:** 1Tissue Engineering for Orthopaedics & Mechanobiology (TOM), Department for BioMedical Research (DBMR), Faculty of Medicine, University of Bern, CH-3008 Bern, Switzerland; xingshuo.zhang@dbmr.unibe.ch (X.Z.); julien.guerrero@dbmr.unibe.ch (J.G.); andreas.croft@dbmr.unibe.ch (A.S.C.); 2Department of Orthopaedic Surgery and Traumatology, Inselspital, Bern University Hospital, University of Bern, CH-3010 Bern, Switzerland; christoph.albers@insel.ch (C.E.A.); sonja.haeckel@insel.ch (S.H.)

**Keywords:** intervertebral disc (IVD), nucleus pulposus progenitor cells (NPPCs), angiopoietin-1 receptor (aka. Tie2), spheroid-formation assay, suspension culture

## Abstract

Lower back pain is a leading cause of disability worldwide. The recovery of nucleus pulposus (NP) progenitor cells (NPPCs) from the intervertebral disc (IVD) holds high promise for future cell therapy. NPPCs are positive for the angiopoietin-1 receptor (Tie2) and possess stemness capacity. However, the limited Tie2+ NPC yield has been a challenge for their use in cell-based therapy for regenerative medicine. In this study, we attempted to expand NPPCs from the whole NP cell population by spheroid-formation assay. Flow cytometry was used to quantify the percentage of NPPCs with Tie2-antibody in human primary NP cells (NPCs). Cell proliferation was assessed using the population doublings level (PDL) measurement. Synthesis and presence of extracellular matrix (ECM) from NPC spheroids were confirmed by quantitative Polymerase Chain Reaction (qPCR), immunostaining, and microscopy. Compared with monolayer, the spheroid-formation assay enriched the percentage of Tie2+ in NPCs’ population from ~10% to ~36%. Moreover, the spheroid-formation assay also inhibited the proliferation of the Tie2- NPCs with nearly no PDL. After one additional passage (P) using the spheroid-formation assay, NPC spheroids presented a Tie2+ percentage even further by ~10% in the NPC population. Our study concludes that the use of a spheroid culture system could be successfully applied to the culture and expansion of tissue-specific progenitors.

## 1. Introduction

The disability induced by lower back pain (LBP) is a global issue with a substantial social burden [1]. The main reason for LBP, apart from trauma to the intervertebral disc (IVD), is the degeneration of the intervertebral disc (IVDD) caused by aging or genetic predisposition [2]. Disability induced by IVDD means a reduction in the labor force participation rate and a tremendous decrease in life quality for patients. However, the precise mechanism of IVDD is still not clear [3].

The IVD consists of three central tissues: (i) the nucleus pulposus (NP), (ii) the annulus fibrosus (AF), and (iii) the cartilaginous endplate (CEP) [4]. The NP located in the center of the IVD is a gelatinous tissue surrounded and kept in place by a criss-cross laminar fibre composing the AF tissue [4]. The health of the IVD depends on the well-being of all three tissue components [5]. If only one of the three tissues partly degenerates, it will affect the health status of the entire IVD.

The NP cell (NPC) population was defined as heterogeneous because of the presence of chondrocyte-like “nucleopulpocytes” and notochordal cells (NCs) within the NP tissue, and possibly other yet unidentified cell populations [6,7,8,9]. It is known that these NCs become scarce and finally disappear with the maturation of the IVD during growth [6]. Recently, a novel type of tissue-specific progenitor cell has been identified and characterized in NP tissue [10,11,12]. These so-called tissue-specific nucleus pulposus progenitor cells (NPPCs) [10,11,12] are multipotent and autochthonous. Sakai et al., (2012) mainly detected two main characteristics of NPPC: (i) NPPCs prefer to form a spheroid colony-forming unit (CFU-s) rather than a fibroblastic colony-forming unit (CFU-f) within a CFU-assay in methylcellulose semi-solid medium [10,11,12], and (ii) NPPCs are positive for the cell surface marker angiopoietin-1 receptor (aka. Tie2) [10,11,12]. Additionally, Sakai et al. found that the disaloganglioside 2 (GD2) is negative in dormant NPPCs and positive in proliferating NPPCs [10]. Although the NPPCs do not disappear as drastically in adults as the NCs do; however, their numbers still decrease with age and IVDD [10].

In clinics, the reality is that the amount of NP tissues obtained from patients are limited, both in the number of donors and in the total cell numbers that scientists can extract from available tissues. In this context, a previous study already compared three standard immunology-based sorting techniques—fluorescence-activated cell sorting (FACS), magnetic-activated cell sorting (MACS), and a mesh-based label-free cell sorting system (pluriSelect)—for sorting NPCs Tie2+ on the most profitable method [12]. Although FACS seemed to be the most efficient method, the vast cell loss of all three methods still limited the NPPC harvest [12]. Based on the low density of NPCs in the tissue and on the low percentage of NPPCs within the entire NPC population, the research to characterize the NPPCs and to develop strategies for regenerative IVD therapy remains challenging. Thus, cell expansion protocols for these NPPCs are urgently warranted. In this study, we aimed to expand the heterogenic NPC population and to increase the percentage of Tie2+ NPCs by using non-plastic adherent cultures.

Traditional cell culture for IVD research was first performed in monolayer to expand these cells in higher numbers for subsequent 3D cell culture. Previous studies have been able to demonstrate that the native phenotype can be better maintained by adjusting oxygen concentration and/or osmolality [13,14,15,16]. On the one hand, it was described that monolayer culture changed NPCs’ morphology and changed the content of key components of the extracellular matrix (ECM). Previous research on 2D NPCs culture showed that the gene expression of collagen type 2 (COL2) decreased significantly from passage (P) 1 to P2, and aggrecan (ACAN) decreased significantly from P0 to P1 [17]. Furthermore, 2D NPCs culture revealed that the gene expression change seemed to happen at early P and then tended to stabilize [17,18,19].

Moreover, the monolayer culture of NPCs showed an increased senescence and oxidative stress [18,19]. Previous studies on NPCs achieved better outcomes in cell phenotype and ECM production with 3D culture within a biomaterial scaffold compared to monolayer culture [20,21,22,23]. On the other hand, Tekari et al. (2016) [11] demonstrated that bovine Tie2+ NPCs lose their phenotype after one week of plastic monolayer culture and are outcompeted by Tie2- cells in normoxia. This de-differentiation could be modestly improved by lowering the oxygen content to 2% and by adding growth factors such as fibroblast growth factor 2 (FGF-2) or growth and differentiation factor 5 (GDF-5) [11].

The spheroid-formation culture is more suited for Tie2+ NPCs compared with simple monolayer culture as Tie2+ NPCs in CFU-assays showed a preference for spherical morphology (CFU-s) in methylcellulose semi-solid medium, and not a fibroblastic morphology (CFU-f) [10,11,12]. An assay similar to the CFU-assay is the spheroid-formation assay (also called “colonosphere” assay), which is the formation of spheroid-like solid aggregates of cells in the suspension environment [24,25,26]. Erwin et al. (2013) showed that spheroids derived from canine NPCs in spheroid-formation assays were indeed multipotent [27]. However, in their study, it seemed that a different type of NPPC population had been identified and characterized rather than the one presented in Sakai et al. (2013) [12]. Erwin et al. characterized these based on several genes which are known as “stemness” markers, i.e., octamer-binding transcription factor 4 (*OCT4*), sex-determining region Y-box 2 (*SOX2*), *NANOG*, *PROM1* (the gene responsible for CD133 expression), Nestin and Neural cell adhesion molecule (*NCAM*) [28,29,30].

In this study, human-derived primary NPC culture was tested on three different surfaces: (i) “classic” (non-molecular treated polystyrene), (ii) a gelatin-coated surface for improved attachment, and (iii) an ultra-low attachment surface. On the ultra-low attachment culture, a focus was given only to the spheroid-forming cells; this is known as a spheroid-formation assay [24,25,26]. The argument for enriching the culture by an attachment-improved surface compared to “classic” surface culture is based on the research on other stem cell types, such as neuro stem cells, which prefer to proliferate in suspension culture on an ultra-low attachment surface or in an attachment-improved monolayer culture environment [26,31]. Attachment-improved culture surfaces also play a role in the protection of the pluripotent capacity of stem cells; for example, in induced pluripotent stem cells (iPSCs) [32]. Coating flasks with gelatin has been described as a straightforward way to improve cell adherence.

In this study, we had two aims: the first aim was to investigate whether spheroid-formation assays could increase the proliferation of Tie2+ NPCs in the whole NPC population compared to monolayer culture on “classic” or attachment-improved surfaces. The second aim was to investigate the implication of NPC spheroids formed on an ultra-low-attachment surface.

## 2. Results

### 2.1. Correlation between Tie2+ Cell’s Morphology and the Culture Surface

The NPCs’ morphology differed considerably based on the culture surfaces used in this study. NPCs grew as a monolayer on the classic surface (Control group) as on the 0.1% gelatin-coating group (Gelatin group) (Figure 1a). However, the group that differed the most was the one on the ultra-low attachment surface (Spheroid group), where the NPCs formed spheroids (Figure 1a).

Compared to the control group, the NPCs of the gelatin group were spindle-shaped and mostly polygonal shaped. The circularity of the NPCs (Figure 1b) was significantly higher in the control group compared to the gelatin group (Mann–Whitney U test, *p* < 0.0001) [33,34,35]. We assessed the cell length as the major axis and the cell width as the minor axis. The aspect ratio (major axis/minor axis) of the gelatin group showed that the NPCs cultured on the gelatin-coated surface presented an elongated and stretched morphology compared to the control group grown on “classic” plastic surface [33,34,35] (Figure 1c).

### 2.2. Colony Morphology Formed by NPCs in CFU-Assay

To follow-up on the study on NPPCs (NPCs Tie2+) by Sakai et al. [10], we looked into the potential of resuspended NPCs to form CFU-s (Figure 2a) and CFU-f (Figure 2b) on methylcellulose, as described previously [10,11,12]. Cell colonies with the phenotype of mixed CFU-s and CFU-f were identified as CFU-s/f (Figure 2c). To better visualize the cells’ morphology, the NPCs were stained with calcein acetoxymethyl (calcein AM). We observed that the circularity of CFU-s and CFU-f were significantly different (*p* < 0.0001) (Figure 2d) [33,34,35]. In culture on methylcellulose medium, cell clones were showing a CFU-s/f morphology characterized by two distinct populations in terms of circularity (Figure 2b).

### 2.3. Correlation between Tie2 Positive NPC Proliferation and the Culture Surface

The percentage of NPPCs (Figure 3a) that were Tie2+ NPC reached 10 ± 5% (mean ± SD) in the control group after ten days of culture. In the gelatin group, the percentage of Tie2+ cells was 4 ± 5% and 36 ± 16% (mean ± SD) in the spheroid group (Figure 3a). The Tie2+ NPC yield of the spheroid NPCs in the spheroid group increased significantly (more than three-fold) compared to the monolayer NPCs in the control group (*p* = 0.0422) and the gelatin group (*p* = 0.0005). Lastly, the percentage of Tie2+ cells in the gelatin group was ~6% lower compared to the control group.

The medians of fluorescence intensity (MFI) of living single cells in the gelatin and spheroid group were compared with the control group (assigned to 1.0) (Figure 3b). The MFI of the gelatin group was significantly decreased compared to the control (*p* = 0.028) (0.7 ± 0.2 fold) (mean ± SD) and the MFI of the spheroid group was slightly increased compared to the control (1.2 ± 0.5 fold).

The population doubling level (PDL) per P (Figure 3c) was calculated to study whether enriching the Tie2+ NPC yield of the spheroid NPCs attributed to either the improved proliferation of Tie2+ NPCs or inhibition of Tie2- NPCs. On the one hand, the PDLs of the spheroid group (6.1 ± 4.5) (mean ± SD) are similar to Tie2+ NPCs in the control group (6.2 ± 3.8) (mean ± SD) (*p* > 0.999, Kruskal–Wallis (K-W) signed-rank test), the gelatin group (3.9 ± 4.2) (mean ± SD) (*p* > 0.999, K-W signed-rank test). On the other hand, the spheroid-formation assay inhibited the PDL of Tie2- NPCs from 3.6 ± 1.2 to 0.8 ± 2.2 (mean ± SD), compared to control (*p* = 0.1547, K-W signed-rank test) and 3.9 ± 1.0 to 0.8 ± 2 (mean ± SD), compared to gelatin groups (*p* = 0.0694, K-W signed-rank test).

The number of clones with cells ≥ 10 cells per 1,000 cells that were seeded in methylcellulose-based medium was counted as CFU-s, CFU-s/f, and CFU-f, respectively (Figure 3d). The number of each clone type on gelatin and the spheroid groups were also compared to the control group (assigned to 1.0). We found that the NPCs from spheroid groups showed a significantly higher capability for CFU-s (*p* = 0.0308) formation compared to the control group.

In the conclusion of the above results, the spheroid-formation assay on an ultra-low attachment surface enriched the Tie2+ NPC yield of the whole population and monolayer culture on a gelatin-coated surface decreased the Tie2+ NPC yield, compared to the monolayer culture on both “classic” surfaces. The proliferation of Tie2- NPCs was almost inhibited in the spheroid-formation assay compared to monolayer culture. As a result, inhibiting the proliferation of Tie2- NPCs enriched the Tie2+ NPC yield of the NPCs of spheroid-formation assay.

### 2.4. Spheroid Reformed after Papain Digestion Increased the Yield of Tie2+

To assess the feasibility of NPCs to be passed from suspension culture to an ultra-low attachment surface, we digested the spheroids and assessed the spheroid-formation assay again. At the same time, we found for the first time the formation of spheroids from NPCs similar to 1st-generation spheroids (Figure 4a). After digesting the 1st-generation spheroids to single cells, and subsequently culturing them on an ultra-low attachment surface, the single-resuspended NPCs were able to reform 2nd-generation spheroids (Figure 4b).

The NPCs suspended from 1st-generation spheroids and 2nd-generation spheroids were assessed for their Tie2+ percentage and CFU-assay. On the one hand, the percentage of Tie2+ NPCs in resuspended NPCs from 1st-generation spheroids NPCs was 30 ± 8.7% (mean ± SD). The percentage of Tie2+ NPCs in resuspended NPCs from the 2nd spheroid was 43 ± 6% (mean ± SD), respectively (Figure 4a). The Tie2+ NPC yield increased by 13% after passaging 1st spheroids and reforming 2nd spheroids (*p* = 0.0152; Figure 3c). On the other hand, the percentage of cells from the 2nd-generation spheroids was 2.2 ± 0.2 (mean ± SD) times higher than CFU-s colonies of the 1st-generation spheroids. The number of CFU-s/f colonies from the 2nd-generation spheroids were significantly decreased compared to the 1st-generation spheroids. Passaging NPCs with the spheroid-formation assay increased the Tie2+NPC’s yield by a factor of ~3.6 (Figure 3a), and the 2nd-generation spheroids had an even higher percentage of Tie2+ NPCs compared to the 1st-generation spheroids.

### 2.5. Extracellular Matrix of NPC Spheroids

The ECM composition of NPC spheroids was assessed by immune-histology. The ECM of the NP mainly consists of ACAN and collagen type 2 COL2; both were confirmed on fixed NPC spheroids (Figure 5). The ECM was mainly distributed between cells composing the spheroids.

### 2.6. Correlation between RNA Expression of NPCs and the Culture Surface

The analysis of the relative gene expression of cells cultured on different surfaces of “classic”, gelatin-coated, and ultra-low attachment for 10 days was performed (Figure 6). The spheroid formation increased the *ACAN* gene expression (*p* = 0.0124) and *COL2* (*p* = 0.2363) expression compared with control group. The *COL1* gene expression in the spheroid (1.9 ± 0.7) and the gelatin group (1.8 ± 2.4) was similar and slightly higher than the control group (assigned to 1.0). The relative gene expression of *NANOG* and *OCT4* within spheroids were the highest among the three groups and significantly higher than the gelatin group (*NANOG*, *p* = 0.0079; *OCT4*, *p* = 0.0352). The relative gene expressions of *TEK* and *KRT19* in the spheroid group were similar to the control group. The result showed that the cells cultured on an attachment-improved surface did not alter in gene expression compared to the cells cultured on “classic” surface. The ECM-related and stemness-related genes tended to be upregulated with the spheroid formation on the suspension environment compared to the cells on monolayer.

## 3. Discussion

### 3.1. The Spheroid-Formation Culture History and Its Implications in Stem Cells

In 1992, Reynolds and Weiss described one of the earliest spheroid-formation assays [36]. They compared the neural stem cells from mice’s striatum cultured or in suspension culture or “classic” attachment culture with media supplemented with epidermal growth factor (EGF). Interestingly, the mice’s striatum cells did not proliferate if cultured as monolayer. However, suspension culture improved the proliferation of cell spheroids but increased the senescence of non spheroid cells. Self-renewal was observed with the single cells from resuspension and reassembly of spheroids. In their study, the cells resuspended from spheroids were able to differentiate into both neurons and glial cells in monolayer culture. Only these resuspended cells could then undergo proliferation in monolayer culture. Additionally, the immunocytochemistry of the neural cells during suspension culture and monolayer culture displayed significant differences. As a result, the authors showed that spheroid formation in suspension culture improved the proliferation of neural stem cells. Spheroid formation was also used to isolate stem cells [36]. Furthermore, spheroid-formation culture has become the mainstream way to culture neural stem cells [26].

The implications of spheroid-formation assay are involved in many different types of stem cells from other tissues. Three important aspects are key in stem cell research: isolation, proliferation, and maintenance of their stemness. The spheroid-formation assay is already widely used in cancer stem cells isolation or enrichment [25,37], lethal screen subtype [37], and the drug-resistant subtype [38]. Two recent studies showed that spheroid formation could protect the multipotency of adipose-derived stem cells [24,39]. The key question is whether spheroid formation is essential for the maintenance of Tie2+ NPPCs.

### 3.2. The Effects of Spheroid-Formation Culture in NPCs

Previous, studies have confirmed that 3D cell culture is beneficial for the phenotype of “nucleopulpocytes” [7,40,41]. Recently, niche biomaterials were developed for nucleopulpocytes that allow maintaining the progenitor status of cells in general by incorporating laminin [42]. The suspension culture in this research showed that the spheroid assay inhibited the proliferation of Tie2− NPCs, and as a consequence purified the whole NPC population closer to a consistent Tie2+ NPC population. In terms of preserving the cell’s stemness, the yield of NPCs Tie2+ in spheroid culture was about three-fold higher than in monolayer cells cultured on the “classic” surface. Moreover, NPCs from spheroids in the suspension-culture on ultra-low attachment surfaces formed about three times more CFU-s colonies than spheroids from monolayer on “classic” surfaces. However, the Tie2+ NPCs’ PDL did not increase compared to monolayer culture on both “classic” and attachment-improved surfaces. As the aim of expanding NPCs Tie2+ from primary NPCs before the sorting of NPCs Tie2+, the spheroid assay decreases the NPCs Tie2−, this will reduce the sorting time cost and the stress to Tie2+ cells caused by FACS sorting.

### 3.3. Can NPPCs Be Used in Cell Transplantation?

We hypothesized that the fibrotic phenotype of NP on IVDD could partially be caused by an increase of fibroblastic cells, which formed CFU-f in these primary cultures. One characteristic of degenerated NP tissues is that these become more fibrotic compared to young and healthier tissue. From the observation of the morphology of the colonies of CFU-assay, there are three different colony types distinguishable: CFU-s and CFU-f, and CFU-s/f with mixed characteristics of both. To recall, both CFU-s and CFU-f types are most likely derived from NPC Tie2+. The formation of CFU-f in purified Tie2+cells supported this hypothesis [12]. To confirm this, Tie2 lineage analysis should be performed in further research. CFU-f differentiated into the non-NPC phenotype. Sakai et al., (2013) [8] stated that CFU-f’s are derived from non-NP phenotypes. We can only speculate here that the NP tissue fibrosis observed in IVDD could be related to the transition from progenitors into CFU-f phenotype. The CFU-f phenotype population differentiated from Tie2+ NPCs might express Tie2 (aka. TEK gene) at a very low level compared to the NP phenotype cells, but the confirmation of this hypothesis should by further tested by RNA in situ hybridization. This could explain why spheroid NPCs with higher percentages of Tie2+ NPCs have a higher CFU-s formation ability. Still, at the whole NPCs population-level have similar MFI of Tie2+ antibody and *TEK* gene expression levels compared with the classic monolayer culture.

We expected that the transplantation of NPCs Tie2+ could be used for the regeneration of IVDs. Further research should investigate the CFU-f phenotype population and the role of CFU-f population in IVDD. Before the mechanism is not understood why CFU-s might transform into CFU-f from NPPCs, the cell therapy with injecting of Tie2+ NPCs is possibly far from being applicable to the clinics.

To establish a tissue transplant donor, we need to identify the cells and their respective ECM. NPC spheroids lack ECM support conversely to the traditional 3D culture of NPCs within a scaffold. The lack of similarity to any in vivo environment and absence of ECM will reduce the proliferation rate in a majority of cell types [43]. Based on the research of Sakai et al. (2012), the CFU-s colonies formed with Tie2+ NPCs in methylcellulose semi-solid media could spontaneously generate ECM with collagen type 2 and aggrecan. With the expression of *ACAN* and collagen type 2 (*COL2*), NPCs spheroids showed a relatively physiological phenotype compared to monolayer on plastics.

## 4. Materials and Methods

### 4.1. NPC Isolation

Human NP tissues from patients undergoing spinal surgery were obtained with approval from the local ethical committee (Ref. 2019-00097). All of the patients provided written consent. The donor details are shown in Table 1 The distinction of NP, CEP, and AF tissues was performed by experienced spine surgeons, i.e., by C.E.A. and S.H. However, for this study, only NP tissues were included. The cells isolation process followed a two-step digestion protocol with 1.9 mg/mL pronase (Roche, Basel, Switzerland) for one hour and followed by 65 U/mL collagenase II (Worthington, London, UK) overnight. The clearance of debris was done by filtration through a 100 µm cell strainer (Falcon; Becton & Dickinson, Allschwil, Switzerland). Total cell numbers and viability were assessed on the next day by using the Neubauer-improved chamber with trypan blue.

### 4.2. NPCs Seeding on Different Surfaces

To study the impact of different culture surfaces on the proliferation of Tie2+ NPCs in the whole NPC population, three different culture surfaces were studied: (i) the “classic” cell culture plastic that was used as a control group, in comparison to (ii) the attachment-improved surface with 0.1% gelatin-coating (gelatin group), and (iii) the ultra-low-attachment culture plastic (=spheroid group). NPCs from the control and gelatin groups were cultured in monolayer, and the whole NPC population of these two groups was assessed for Tie2+ NPC yield and CFU assay. In the spheroid group, only the spheroid-NPC population was harvested and assessed for Tie2+ NPC yield and CFU-assay. The experimental design is shown in Figure 7. The “classic” surface flask was from TPP™ (#cat 90076, Trasadingen, Switzerland). The culture surface of the “classic” control group has a non-molecular treated polystyrene surface with optomechanical treatment. The 0.1% gelatin-coated flask was prepared at least one day before seeding. Coating gelatin (cat# 04055, Fluka, a sub-brand of Sigma) from porcine skin was used. The gelatin coating protocol was followed according to the spheroid research by Redondo-Castro et al. (2018) [44]. The gelatin was dissolved in distilled water overnight. Afterwards, the 0.1% gelatin was then autoclaved and stored at 4 °C and was used within two weeks. Before coating, the 0.1% gelatin was warmed in a water bath at 37 °C. The flasks were coated with 0.1% gelatin overnight and washed with Phosphate-buffered saline (PBS). The ultra-low attachment surface flask was from Corning (#cat 3814 Wiesbaden, Germany), and ~200,000 NPCs were seeded into each flask. The culture media was low glucose Dulbecco’s Modified Eagle Medium (LG-DMEM) (#cat 11-885-084 Gibco, Life Technologies, Basel, Switzerland) media with 10% FCS (cat# F7524, Sigma-Aldrich, Buchs, Switzerland) and 1% Penicillin/Streptomycin/Glutamine (P/S/G) (cat# 10378016 Gibco) supplemented with 2.5 ng/mL fibroblast growth factor 2(FGF-2) (Peprotech, UK, London) to maintain the progenitor phenotype [44]. 

### 4.3. NPC Freezing and Thawing

NPCs were expanded up to P1 on polystyrene flask T150 or T300 (TPP™, VWR, Dietikon Switzerland) with LG-DMEM media with 10% FBS and 1% P/S/G. The NPCs were resuspended in 10 mL Trypsin-EDTA Solution (0.25%) (1×) for 5 min. The enzyme digestion was stopped by adding 10 mL of culture media. NPCs were washed with 10mL cold PBS by centrifugation (500× *g* for 5 min). The total cell count was performed with a Neubauer-improved hemocytometer. One million NP cells/mL were suspended in freezing media, and frozen at - 80 °C in a freezing container (“Mr. Frosty” cat#5100-0001; Thermo Fisher Scientific, Waltham, Massachusetts, USA) filled with 2-propanol. The freezing media consisted of the same culture media and supplemented with 10% Dimethyl sulfoxide (DMSO). After two days, the frozen NPCs were transferred from −80 °C to −150 °C for long term storage. Prior to the experiments, NPCs were quickly thawed at 37 °C and were then washed with 10 mL 4 °C-cold PBS by centrifugation (500× *g* for 5 min) and counted before seeding.

### 4.4. NPC Digestion and Spheroid Isolation

The single cells were removed with a 30 µm cell strainer (cat# 43-50030-03, pluriStrainer^®^, pluriSelect Life Science, Leipzig, Germany). The spheroid-like NPCs were cultured on an ultra-low attachment surface, and the monolayer NPCs were cultured on the “classic” surface and gelatin-coated (attachment-improved) surface. NPCs on these three types of surfaces were resuspended with the same digestion media according to the protocol of the spheroid research by Koito et al. [45]. The digestion media consisted of Hank’s Balanced Salt Solution (HBSS; lacking Ca^2+^, Mg^2+^) buffer with 10 mM HEPES (N-2-HydroxyEthylPiperazine-N-2-Ethane Sulfonic acid) buffer solution (cat# 15630, Gibco, Thermo Fisher Scientific Inc., Basel, Switzerland), 0.8 mg/mL L-cysteine (cat# 20119, Fluka) and 20 I.U./mL papain (cat# P3125, Sigma). HBSS buffer was then further diluted in PBS with 9.5 mg/mL HBSS (cat# H2387, Sigma), 35mg/mL NaHCO_3_ (cat# 31437, Sigma) and 1% P/S/G. L-Cysteine (Fluka) was added freshly and adjusted to pH 7.4, after adding the papain (Sigma). To test the efficiency of different digestion enzymes, equal numbers of spheroids were transferred to 6-well plates (cat# 92006, TPP™) after incubation in papain solution ~2 mL (20 I.U./mL) for 5 min.

The total cell count was performed with a Neubauer-improved hemocytometer (cat# 0640110, Marienfeld, Karlsruhe, Germany). The population growth was estimated by the cumulative population doubling level (PDL) according to Formula (1):(1)$$PDL\;=\;\log\left(\frac{N_{t}}{N_{0}}\right)/log2$$
where *N*_0_ = initially seeded cell number and *N_t_* = cell number at time point *t*.

### 4.5. Tie2+ NPC Characterization by Flow Cytometry and Colony Formation Assay

In this study, the preferred expansion method was quantified by measuring the percentage of Tie2+ NPCs (Tie2+) within the NPC population, and the number of colony-forming units (CFUs) per 1000 NPCs [10].

The NPPC surface marker was stained with the Tie2-PE antibody (1:20) (cat# FAB3131P R&D, Muttenz Switzerland). Dead cells were stained with 4′,6-diamidino-2-phenylindole (DAPI) at a concentration of 0.1 µg/mL. The flow cytometry (CytoFLEX S 5350, Indiana, United States) was used to analyze and quantify the Tie2+ NPC yield. The flow cytometry data were analyzed with FlowJo software, version 10.4.2 for Mac OS X (Treestar, Ashland, Oregon, USA) (Appendix A).

Next, 1,000 single NPCs were seeded in 1 mL of a methyl-cellulose-based medium (STEMCELL Technologies, Inc., Grenoble, France) in a 35 mm Petri dish (TPP, VWR). The colonies then formed. Those colonies that were more than or equal to ten cells in total were counted (by X.Z.) under a light microscope (Leica DM IL, Leipzig, Germany).

### 4.6. Immunofluorescence Staining

The NPC spheroid’s extracellular matrix was stained on frozen cryosections. After isolation of spheroids by 30 µm cell strainers (pluriSelect), the spheroids were collected on the bottom of 50 mL Falcon centrifuge tube (cat#21008-178; VWR) with centrifuge at 500 g for 5 min. The spheroids were moved into 1.5 mL Eppendorf tubes (cat#21008-103; VWR) and centrifuge at 500× *g* for 5 min to remove media. The spheroids were moved into the tube with Optimal Cutting Temperature (O.C.T.) compound, #cat 4583, Sakura, Alphen aan den Rijn, Netherlands). After freezing in liquid nitrogen, the frozen spheroids were stored at −150 °C. The frozen sections were cut using a cryostat (Microm HM 560, Thermo Fisher Scientific). The frozen section was then fixed by 4% paraformaldehyde (PFA) for 20 min at room temperature. The sections were extensively washed with Tris-buffered saline (TBS) (#cat 15504-020, Thermo Fisher Scientific) containing 0.025% *v*/*v* Tween 20 (#cat P1379, Sigma). To prevent unspecific staining the slides were then covered by blocking buffer (3% goat serum TBS) for 30 min at RT. Then, the primary antibodies of aggrecan (ACAN) (1:50) (cat# TA336492, Origene, Muttenz, Switzerland), and of type two collagen (each at a 1:50 dilution) (cat# sc-7764, Santa Cruz Biotechnology, Inc., Santa Cruz, California, USA) were incubated overnight at 4 °C. After extensive washing with TBS added 0.025% *v*/*v* Tween 20 (= TBS-Tween20), the secondary antibodies (both at concentrations of 1:200 diluted) Alexa Fluor^®^ 555 Rabbit Anti-Goat IgG (H+L) (cat# 1722388), and Alexa Fluor 488 Goat anti-Rabbit SFX Kit (cat# A-11008) (both from Thermo Fisher Scientific, Inc., Reinach, Switzerland) were incubated for three hours at room temperature in the dark. After extensive washing with TBS-Tween20, the slides were mounted in DAPI containing glycerol-based embedding medium (#cat ab104139, Abcam, Cambridge, UK). The ECM fluorescence image was taken with a Nikon Eclipse 800 microscope (Nikon, Tokyo, Japan).

### 4.7. Morphology of Cells

To better visualize the morphology of cells, live cells were pre-stained with 2μM Calcein-AM for 10 min at 37 °C. Cell’s morphology was then quantified by ImageJ v1.53c on the microscopic pictures using “cell body circularity” [33,46]. An example of how the cells outlines were defined by manual segmentation can be found in the Appendix A.

### 4.8. qPCR

Three groups of RNA expressions of human genes were confirmed of the NPCs within three types of culture surfaces and the cells before seeding (day 0). The first group consists of the ECM related genes, i.e., aggrecan (*ACAN)*, collagen type 1 (*COL1*), and collagen type 2 (*COL2*). The second group has genes related to IVD phenotype (*KRT19)* and NPPCs phenotype (*TEK* for Tie2). The third group consists of the genes that are related to cell stemness (*Nanog* and *Oct4)*.

To assess whether the monolayer culture will influence the phenotype of NPCs, we compared the relative gene expression between the cells one day 0 before seeding and at day ten of the monolayer culture on a “classic” surface (control group).

The RNA of spheroid NPCs was extracted with the NucleoSpin RNA XS kit followed by DNA digestion (cat# 740902.10/.50/.250, Machinerey-Nagel, Oensingen, Switzerland). The RNA from the cells cultured on a monolayer was first resuspended in 20 I.U./mL papain (cat# P3125, Sigma) in Hank’s Balanced Salt Solution (HBSS; lacking Ca2+, Mg2+) buffer with 10 mM HEPES (N-2-HydroxyEthylPiperazine-N-2-Ethane Sulfonic acid) buffer solution (cat# 15630, Gibco, Thermo Fisher Scientific Inc., Basel, Switzerland), 0.8 mg/mL L-cysteine (cat# 20119, Fluka) and 20 I.U./mL papain (cat# P3125, Sigma), and then the RNA was extracted with the GenElute™ Mammalian total RNA purification kit including DNA digestion (cat# RNB100 Sigma-Aldrich, Buchs, Switzerland). The RNA of all samples was assessed with reverse transcription with high capacity cDNA Kit (cat# 4368814, Thermo Fisher). The qPCR was performed in duplicates (CFX96 Touch, Bio-Rad, Cressier, Switzerland) with mixed cDNA and (250 nM) forward and reverse primers (Microsynth, Balgach, Switzerland) in iTaq™ universal SYBR^®^ Green Supermix (Bio-Rad). The relative gene expression was calculated using the 2^−ΔΔCq^ method with two reference genes, i.e., 18S and GAPDH, respectively. The data were normalized to the control group, which were cells cultured on “classic” surface flasks. The primers can be seen in Table 2.

### 4.9. Statistics

Statistical analysis was performed using Prism 7.0d for Mac OS X (GraphPad, La Jolla, CA, USA). Statistical significance was determined using the Mann–Whitney U test (two groups) and Kruskal–Wallis (K-W) signed-rank test followed by a Dunn’s multiple comparison test (three groups). Values are given as means ± SD. A *p*-value <0.05 was considered to be significant.

## 5. Conclusions


This study showed that the expansion of NPCs in the 2D monolayer is not favorable for the Tie2+ NPC phenotype.The gelatin modification did not show any advantage for Tie2+ NPC proliferation compared with the control and spheroid group. The attachment-improved surface did not show the improvement of cultured whole NPCs population compared with standard plastic surface.Spheroid-formation assay enriched the Tie2+ population of NPCs from ~10% on monolayer culture to ~36% in suspension culture.Generation of 2nd generation spheroid-formation assay by reassembly of NPC spheroids increased the Tie2+ NPC population yield even further from about ~30% to ~43%.We introduced a new culture protocol in which Tie2+ NPCs can be maintained and Tie2- NPCs can be inhibited in suspension culture.NPC spheroids resulted in more purified Tie2+ NPCs compared to monolayer. However, more research would be required to see if the Tie2+ population could be even further purified with increased passaging into 3rd or even 4th-generation spheroids.


## Figures and Tables

**Figure 1 ijms-21-09423-f001:**
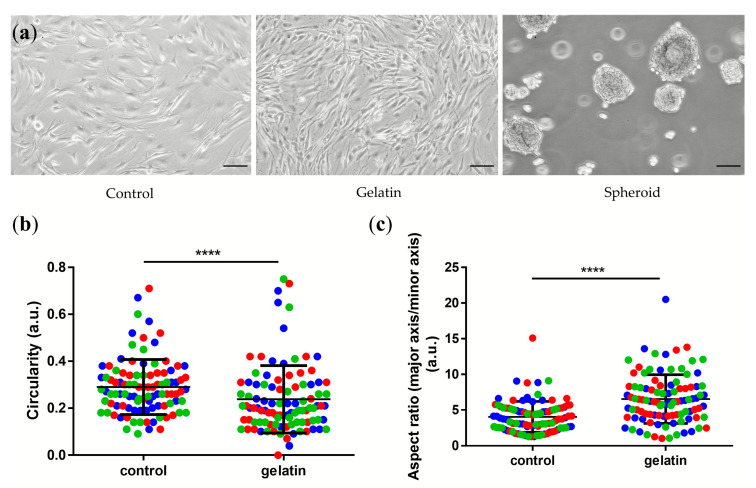
(**a**) Phase-contrast microscopy pictures of NPCs grown on three different surfaces. Scale bar = 100 µm. Here, “Control” represents cells cultured in “classic” T75 flasks, “Gelatin” refers to cells cultured in 0.1% gelatin-coated T75 flasks, and “Spheroid” refers to cells from the spheroid formed assay in ultra-low attachment T75 flasks; “a.u.” refers to arbitrary units. Scale bar = 100 µm. (**b**) Circularity and (**c**) aspect ratio (major axis/minor axis) was measured using imageJ software between control and gelatin groups. In (**b**,**c**), each dot represents one cell (n = 100), taken from three different donors and marked in red, green, and blue for both groups respectively, Mann–Whitney U test, **** = *p* < 0.0001. Lines represent means ± standard deviation (SD).

**Figure 2 ijms-21-09423-f002:**
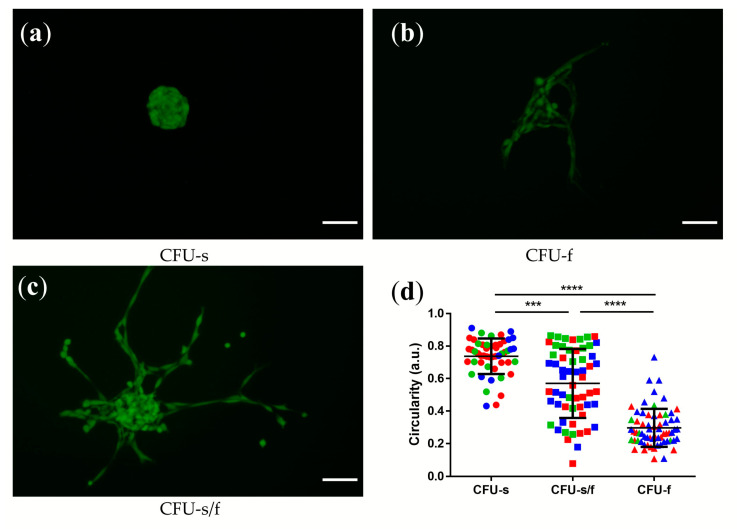
Phase-contrast microscopy images (10x) of three types of colony-forming units (CFU): (**a**) spheroid-type (CFU-s), (**b**) fibroblastic type (CFU-f), and (**c**) semi-spheroid-and-fibroblastic type (CFU-s/f); the NPCs were stained with Calcein-AM; scale bar = 100 µm (**d**) Cell’s circularity of different types of clones; “a.u.” refers to arbitrary units. Each dot represents one cell (n = 120), taken from three donors of patients marked in red, green, blue; Kruskal–Wallis (K-W) signed rank test, *p* = 0.0013 (CFU-s vs. CFU-s/f), *** = *p*-value < 0.001, **** = *p*-value < 0.0001. Lines in (**d**) represent means ± standard deviation (SD).

**Figure 3 ijms-21-09423-f003:**
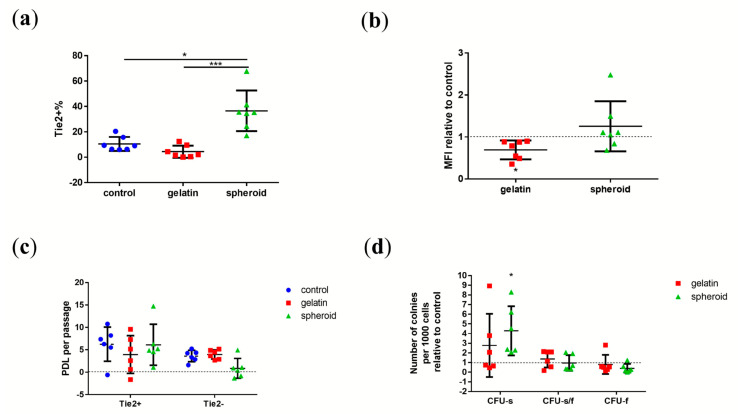
Plot of individual values of Tie2+ cells’ yield in the human NPC population. (**a**) Tie2-PE median of fluorescence intensity (MFI) of living single NPCs relative to control; (**b**) population doubling level (PDL) of Tie2+ NPCs and Tie2- NPCs; (**c**) and quantification of number of CFUs with resuspended NPCs relative to control in different flask types, (**d**) Number of clones with cells ≥ 10 cells per 1000 cells seeded in methycellulose medium after ten days. Here, “control” represents cells cultured in “standard” T75 flasks, “gelatin” represents cells cultured in 0.1% gelatin-coated T75 flasks, and “spheroid” represents cells from the spheroid forming assay in ultra-low attachment T75 flasks, “number of clones relative to control” represents the number of clones formed per 1000 cells of the cells cultured in gelatin/spheroid group relative to control group; N (donors) = 7 in (**a**,**b**), and N = 6 in (**c**,**d**). Kruskal–Wallis signed-rank test with Dunn’s multiple comparison test. *p* = 0.0422 (a, control vs. spheroid), 0.0005 (a, gelation vs. spheroid), 0.028 (**b**), 0.0308 (**d**). * = *p*-value < 0.05, *** = *p*-value < 0.001. Lines in (**a**–**d**) represent means ± standard deviation (SD).

**Figure 4 ijms-21-09423-f004:**
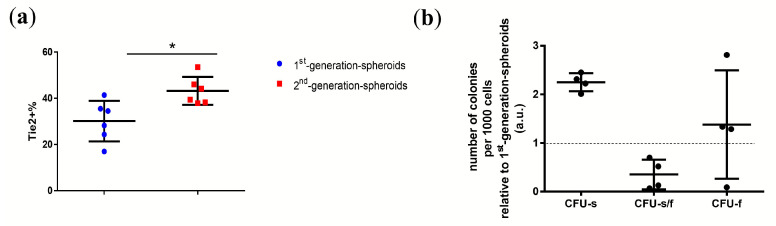
(**a**) Percentages of Tie2+ cell’s in whole human NPC population of 1st-generation spheroids and 2nd-generation spheroids; N (donors) = 6, Mann–Whitney U test, *p* = 0.0152. * = *p*-value < 0.05. (**b**) CFU-assay of NPCs suspended from 1st-generation spheroids and 2nd-generation spheroids, the number of clones per 1000 cells of 2nd-generation spheroids was expressed relative to 1st-generation spheroids. CFU-s stands for spheroid-like shape colonies, CFU-f stands for plastic-adherent fibroblast-like colonies, CFU-s/f represents colonies with a phenotype of mixed characteristics from CFU-s and CFU-f; “a.u.” refers to arbitrary units; N (donors) = 4. Lines in (**a**,**b**) represent means ± standard deviation (SD).

**Figure 5 ijms-21-09423-f005:**
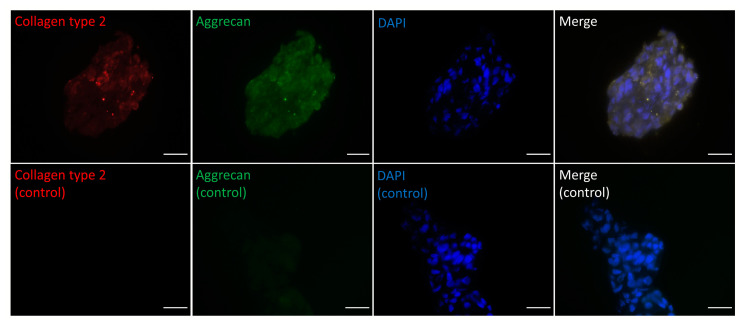
Fluorescent microscopy images of NPC spheroid extracellular matrix (ECM) stained for immune-histology: Collagen type 2 (red), aggrecan (ACAN) (green), and 4′,6-diamidino-2-phenylindole (DAPI) (blue). Merge represents the color merge of red, green, and blue channels. Control represents the absence of primary antibody and secondary antibody control with DAPI (blue). Scale bar = 50 µm.

**Figure 6 ijms-21-09423-f006:**
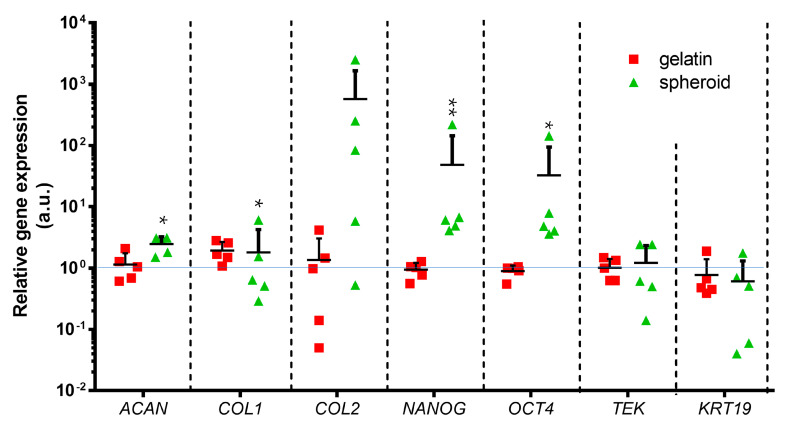
qPCR of genes of cells cultured for ten days on different surfaces. Relative gene expression was normalized to the control (relative assigned to 1.0), which was “classic” T75 flask culture for ten days. Figure legend: “gelatin” represents cells cultured in 0.1% gelatin-coated T75 flasks, and “spheroid” represents cells from the spheroid formed assay in ultra-low attachment T75. *ACAN* stands for aggrecan, *COL1* represents collagen type 1, *COL2* stands for collagen type 2, *NANOG* stands for Homeobox transcription factor Nanog, *OCT4* stands for octamer-binding transcription factor, *TEK* stands for TEK receptor tyrosine kinase (Tie2), and *KRT19* stands for Keratin 19; “a.u.” refers to arbitrary units; N (donors) = 5. Kruskal–Wallis signed-rank test with Dunn’s multiple comparison test. *p* = 0.0124 (*ACAN*), 0.0079 (*NANOG*), 0.0352 (*OCT4*). * = *p*-value < 0.05, ** = *p*-value < 0.01. Lines represent means + standard deviation (SD).

**Figure 7 ijms-21-09423-f007:**
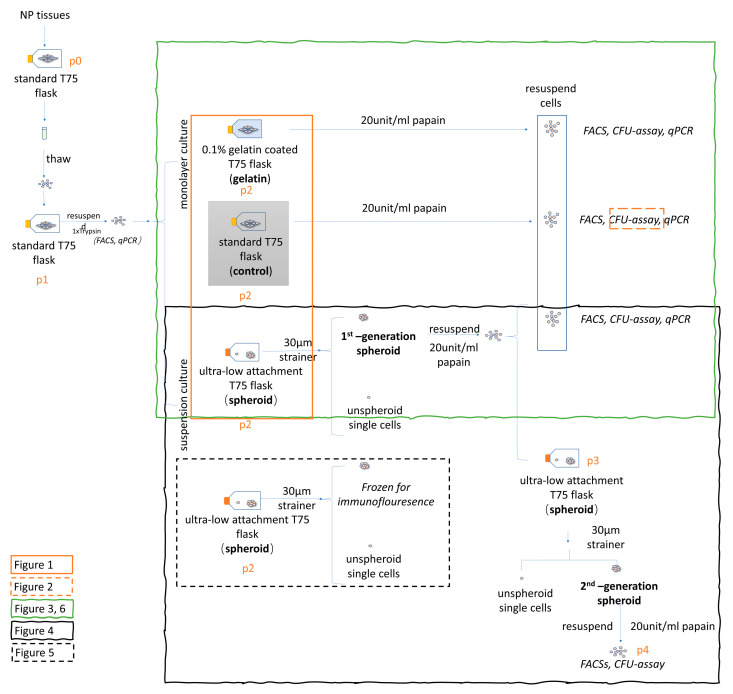
Flow graph of experimental design, important steps are outlined in bold. Figure legend: “p” represents passage.

**Table 1 ijms-21-09423-t001:** Donor list with of gender, age, Pfirrmann grade, tissue weight, and the number of isolated cells.

No.	Gender	Age	Pfirrmann Grade *	Location	Tissue Wet Weight (g)	Number of Isolated Cells
1	female	37	1	T12/L1	0.39	125,000
2	male	39	1	T11/T12	1.58	500,000
3	male	19	1	T12/L1	0.6	200,000
4	male	28	1	L4/L5	1.5	200,000
5	male	28	1	L1/2	1.29	500,000
6	male	40	1	L1/L2	1.33	180,000
7	male	40	2	T11/T1	2.14	500,000
8	male	23	1-2	T11/12	0.63	50000

* The grading was performed by two experienced spine surgeons, i.e., by C.E.A. and S.H.

**Table 2 ijms-21-09423-t002:** Primers list.

Name.	Description	Primer Forward	Primer Reverse
*18S*	Ribosomal 18s RNA gene	CGA TGC GGC GGC GTT ATT C	TCT GTC AAT CCT GTC CGT GTC C
*GAPDH*	Glyceraldenyde-3-phosphate dehydrogenase	ATC TTC CAG GAG CGA GAT	GGA GGC ATT GCT GAT GAT
*ACAN*	Aggrecan core protien	CAT CAC TGC AGC TGT CAC	AGC AGC ACT ACC TCC TTC
*COL1*	Collegen type 1	GTG GCA GTG ATG GAA GTG	CAC CAG TAA GGC CGT TTG
*COL2*	Collegen type 2	AGC AGC AAG AGC AAG GAG AA	GTA GGA AGG TCA TCT GGA
*KRT19*	Keratin 19	TGT GTC CTC GTC CTC CTC	GCG GAT CTT CAC CTC TAG C
*TEK*	TEK receptor tyrosine kinase	TTA GCC AGC TTA GTT CTC TGT GG	AGC ATC AGA TAC AAG AGG TAG GG
*NANOG*	homeobox protein NANOG	AGA ACT CTC CAA CAT CCT GAA CCT	CCT GCG TCA CAC CAT TGC TAT
*OCT4*	octamer-binding transcription factor 4	GAG AGG CAA CCT GGA GAA TT	CCA CAC TCG GAC CAC ATC

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
