# Peer review of "Spheroid-Like Cultures for Expanding Angiopoietin Receptor-1 (aka. Tie2) Positive Cells from the Human Intervertebral Disc"

_ijms, 2020, doi:10.3390/ijms21249423_

Round 1

Reviewer 1 Report

The present version of the manuscript „Spheroid-like Cultures for Cell Expansion of Angiopoietin Receptor-1 (aka Tie2) positive cells from human intervertebral disc” of Zhang et al benefited from extensive revision. Most of the concerns raised by the reviewer have been addressed (e.g. increasing sample number). Further, inclusion of additional data (qPCR analysing ECM expression and stemness) greatly improved the scientific soundness of the manuscript.

However, I have some minor points that should be addressed prior publication:

  1. L108: Correlation between Tie2+ cell’s morphology and the culture surface. This paragraph describes the morphology of NPCs (L109) on different surfaces, in the Figure legend it is stated that growth of NPPCs (L114) was analysed. As I understand, Tie2+ NPPCs are a subpopulation of NPCs. This is especially misleading, since paragraph 2.2 analyses the fraction of Tie2+ cells in correlation to the respective culture surface (Figure 3A). Please clarify which cells were analysed here (NPC or NPPC), and in case of NPPC, how Tie2+ cells were collected and enriched from a NPC population beforehand.
  2. Figure 2 shows the potential of NPCs to form CFU-s and CFU-f depending on the surface they have been grown on before (L131-132). On which surface have the cells been grown that are shown in Figure 2? There seem to be general differences in the ability to form CFU-s/ CFU-f, but it is not shown how the different culturing methods impact colony forming ability.
  3. Figure 6. Mann-Whitney-U test in not the appropriate method for multiple comparison, as authors correctly state in methods section. Please correct. Further, it would improve legibility when abbreviations of the genes are included in figure legend.
  4. Conclusion drawn in L247 contradicts the following statement. Please re-write for clarity.
  5. In discussion section L333-337 authors state that ACAN/CLO2 ratio can be used to distinguish healthy from inflamed NP tissue. How should the ratio for healthy tissue be and, although artificial, does the actual ACAN/COL2 ratio measured in qPCR reflect a healthy (or inflamed) tissue situation?
  6. Spelling/ incomplete sentence:
    1. L135,
    2. Figure 4d is not referenced in text
    3. L210 should reference to Figure 4c (not Figure 3c)?
    4. Figure 4c X-axis labeling is missing

Author Response

Author replies to the queries of reviewers

Dear editors, dear reviewers,

We would like to thank the two reviewers for their time and valuable comments to improve our manuscript. We also would like to thank the editor for providing us with the possibility of submitting a revision.

We have now thoughtfully edited the manuscript and tried to incorporate the majority of the suggestions. We have numbered the points of the reviewers, and the changes in response to Reviewer 1 were highlighted in blue, and the changes of Reviewer 2 are highlighted in orange, and general edits that were done are highlighted in green the new version.

We hope that this new and much-improved version can now be deemed acceptable for publication.

The current version comprises a total 6,140 words, 7 figures, 2 Tables, and 47 references.

Reviewer 1

1. L108: Correlation between Tie2+ cell’s morphology and the culture surface. This paragraph describes the morphology of NPCs (L109) on different surfaces, in the Figure legend, it is stated that the growth of NPPCs (L114) was analysed. As I understand, Tie2+ NPPCs are a subpopulation of NPCs. This is especially misleading since paragraph 2.2 analyses the fraction of Tie2+ cells in correlation to the respective culture surface (Figure 3A). Please clarify which cells were analysed here (NPC or NPPC), and in case of NPPC, how Tie2+ cells were collected and enriched from an NPC population beforehand.

Answer: We apologize for the spelling mistake on L114, we corrected it as NPCs. We did not use a pure population of Tie2+ cells in this research. As we got a limited amount of NP tissues from surgery, a phase of expansion for NPCs was needed before isolation of the NPPCs. The expansion of NPCs was performed on monolayer as described in previous research from our group (Frauchiger, et al. Fluorescence-Activated Cell Sorting Is More Potent to Fish Intervertebral Disk Progenitor Cells Than Magnetic and Beads-Based Methods. Tissue Engineering Part C-Methods, 2019) , and the collection of Tie2+ cells was performed by FACS sorting using a Tie2 antibody conjugated to a fluorescent dye. The collection method is already described in a previous study published by our group (Frauchiger, et al. Fluorescence-Activated Cell Sorting Is More Potent to Fish Intervertebral Disk Progenitor Cells Than Magnetic and Beads-Based Methods. Tissue Engineering Part C-Methods, 2019), which focused specifically on how Tie2+ cells can be isolated by FACS.

2. Figure 2 shows the potential of NPCs to form CFU-s and CFU-f depending on the surface they have been grown on before (L131-132). On which surface have the cells been grown that are shown in Figure 2? There seem to be general differences in the ability to form CFU-s/ CFU-f, but it is not shown how the different culturing methods impact colony-forming ability.

Answer: Thanks for pointing this out. Figure 2 should not be under within section 2.1; we wrote a new title for section 2.2. The cells of Figure 2 of CFU-assay cultured in methylcellulose, and before CFU-assay, the cells were cultured on “classic” surface. This information was added on L135-137. In Figure 2, we showed the morphology of CFU-s, CFU-f, and CFU-f/s. The impacts of colony-forming ability from the different culturing methods are shown in Figure 3d.

3. Figure 6. Mann-Whitney-U test is not the appropriate method for multiple comparisons, as authors correctly state in the methods section. Please correct. Further, it would improve legibility when abbreviations of the genes are included in the figure legend.

Answer: To improve clarity and avoid misunderstanding, we deleted Figure 6.

4. Conclusion drawn in L247 contradicts the following statement. Please re-write for clarity. Answer: We are not sure to which precise statement the reviewer was pointing to. However, we went through the manuscript again and checked that there should be no conflicting statements.

5. In discussion section L333-337 authors state that ACAN/COL2 ratio can be used to distinguish healthy from inflamed NP tissue. How should the ratio for healthy tissue be and, although artificial, does the actual ACAN/COL2 ratio measured in qPCR reflect a healthy (or inflamed) tissue situation?

Answer: We deleted the discussion concerning the ACAN/COL2 ratio as it was an overstatement.

Reviewer 2 Report

See attached file

Author Response

Reviewer 2

  1. this paper contains numerous spelling and grammatical errors, and some sentences are incomplete (Example: L134/135 or L144/145, L219, L333 and more).

Answer: We apologize for the numerous spelling and grammatical errors. The manuscript was improved according to the reviewer’s comments. We completed sentences on L136, L143/144, and L219. Moreover, L333 was deleted to better improve our document.

  1. Other sentences/paragraphs are confusing or difficult to comprehend (Example: L179-181; L192-193; L227-234).

Answer: We went carefully through the entire manuscript and have addressed the stylistic and grammatical issues. To improve the clarity and avoid misunderstanding, we deleted the sentence/paragraphs on L192-193,L227-234.

  1. Also, the rationale for some of the experiments is not quite clear (Fig 1d, Fig 6).

Answer: We would use Figure 1d to identify the morphology of spheroids and Fig6 to show the 2D culture decreases the ECM gene expression. However, these two points are not close to the theme of this research, which is the correlation between Tie2%/ RNA expression of NPCs and the culture surface. To improve the clarity and avoid misunderstanding, we deleted these experiments.

  1. It is also not clear in how far age, degeneration of the IVD, the gender of a donor or location of the disc might have impacted on the results. No correlation was made between Table1 and the donors used in the experiments. Therefore, the overall quality of the manuscript is not satisfactory for publication at this point.

Answer: We agree the correlation of donor/age/IVD degeneration status etc. is important; however, the size of our sample was relatively small with only 7 data points for a valuable correlation. Furthermore, the incorporation of these suggestions would go beyond the scope of this study. We enclose below the correlation coefficients for spinal level, donor gender, and Pfirrmann grading. None of these showed a statistical significance.

  1. Why is the manuscript written in different colours?

Answer: The different colour present in the manuscript were added to highlight the modifications and changes performed during the revision process, highlighting the changes from two previous reviewers.

  1. Abbreviations are not explained; Example: Calcein-AM – What is AM?

Answer: We thank the reviewer for pointing this out. We explained the abbreviation of Calcein-AM on L140.

  1. If gene names are used, they need to be italicized. Example: L91/92, L312

Answer: We thank the reviewer for pointing this out. We went carefully through the entire manuscript and italicized the gene names.

  1. Previous work on cellular heterogeneity and 2D gene expression levels of cultured IVD cells by others is not cited.

Answer: We cited new references for cellular heterogeneity for 2D gene expression level on L49 and 2D gene expression levels in L75-78.

  1. Figure 1: An example image of how a cell outline was defined for these measurements should be provided in the supplementary material.

Answer: We added this image on how a cell outline was defined on Supplementary Figure 2 and cited it on L423,424.

  1. Figure 1: What is the relevance of Fig1d or the meaning of the range in diameter?

Answer: Mentioning the NPCs’ diameter was a detail we initially decided to include in this study as the spheroid culture of NPCs did not have a lot of reports before. However, the diameter of the spheroids did not have a special meaning in this study, so we deleted this data.

  1. Figure 1: Which measure?

Answer: We clarified this the sentence figure legend on L123.

  1. Figure 1: Donors as in different IVDs or cell lines or patients?

Answer: We have clarified the sample sizes in the figure legends. Donors are from different patients (Table 1).

  1. Figure 1: L121 – each dot represents one cell or spheroid?

Answer: We apologize for this unclarity. To improve the clarity and avoid misunderstanding, we deleted Figure 1d.

  1. Figure 2: No correlation to the surface type is provided

Answer: Thanks for pointing this out. Figure 2 should not under the title of section2.1; we rewrote a new title of section 2.2. More information was added on L131-134. In Figure 2, we showed the morphology of CFU-s, CFU-f, and CFU-f/s. The impacts of colony-forming ability from the different culturing methods are shown on Figure 3d.

  1. Figure 2: Figure labelling is mixed up between 2b and 2c in L131-L133.

Answer: We apologize for this mistake. We corrected the labelling.

  1. Figure 2: L145 – cell or colony?

Answer: We mean cell here.

  1. Figure 3: For what reason and for which marker was the MFI measured – Tie2?

Answer: We measured MFI followed the reviewer’s requirement in the first revision process. The reason to measure MFI was that the Tie2 expression level in Tie2+ cells was too low to show a clear separate population, as shown in Supp Figure 2. We added the information of the marker on L172.

  1. Figure 3: Have the authors excluded the impact of autofluorescence?

Answer: As shown on the S1, we used a PE-isotype control to exclude the impact of autofluorescence.

  1. Figure 3: How come data points in Fig 3d CFU-s are very spread out compared to CFU-f/s or CFU-f?

Answer: We assume this is due to donor variance. The CFU-s cells are defined as progenitor cells, and

3 the proliferation of progenitors is influenced by the environment. The role of CFU-s/f and CFU-f cells is still not clear. The CFU-s formation also depends on the quiet state vs proliferation, differentiated level or senescence of NPPCs. The cells’ senescence is also dependent on the proliferation level, from P0 to P1. We tried to establish a cell line to solve this problem; however, this was not successful.

Nevertheless, we would like to thank your question that gave us a new idea for future research. This idea could be used to identify the cells' senescence with beta-galactosidase and to analyze the correlation between Tie2+ cells' CFU-s formation ability and senescence.

  1. Figure 4: Spheroid size between a and b looks very different. A size/diameter comparison should be included here.

Answer: The spheroids shown in Figure 4a, b were randomly chosen. The size range of 1st spheroid was a diameter between 18 and 207 µm, the size range of 2nd spheroid was 44-167 µm (data not shown). To improve the clarity and avoid misunderstanding, we deleted these two pictures.

  1. Figure 4: 4d: Sample number seems quite low, and CFU-f has a large SD, is there an

explanation for that? Difference between CFU-s/CFU-f/s and CFU-f does not appear

to be statistically significant. What do the different colours represent?

Answer:  We agree that the sample number of Figure 4d is a weak point of this study. We lost some donors during medium change because spheroids were suspended in the medium. The CFU-s with their small SD seemed more important for our study than the CFU-f cells. It was the CFU-s cells that were identified as progenitor nucleus pulposus cells. However, the role of CFU-f is not really clear until now. The fibroblastic CFU-f cells have clonality and could not produce ECM of NP tissues (Sakai et al. Nature communications 2012, 3, 1264.). All the characteristics from the CFU-f cells differ from the NP phenotype. The number of CFU-f might be a reflection of the differentiation level of NPPCs. With your comments on Tie2 lineage analysis, we will verify this hypothesis in further research. We apologize for the different colours, and the picture has been simplified to black and white.

  1. Figure 5: There appears to be some background on the green channel.

Answer:  Yes, we agree that there is some background on the green channel. However, it should still be possible to distinguish between the sample group and the control group.

  1. Figure 6: What does N=5 represent, replicates or individual experiments?

Answer: We deleted the experiments without a clear rationale. For the new Fig 6, N=5 means donors from different patients, and the replicates is 2 for each donor (method L443).

  1. Figure 7: How come the data points for COL2 expression and the data points for all gene expression in the spheroids are so spread out?

Answer: The first reason is due to donor variance. The second reason is that the ECM distribution of COL2 of NP tissue isn’t evenly. The NP tissues from the surgeons are from different parts of the NP.

  1. Figure 8: This figure would be helpful to explain the experiment in 2.5.

Answer: We added the Figure number in Figure 8 (now Figure 7) to explain it more clearly.

  1. Results: The information regarding the types of cells used is not sufficient. It is not obvious if the cells used are primary isolates or thawed cell lines. What passage number was used? Are the different cells from different donors of comparable passage number? What about the donor/age/IVD degeneration status etc. in correlation to the presented results?

Answer: We added the Figure number, passage number in Figure 8 (now Figure 7) to explain this more clearly. Yes, the different cells from different donors are comparable as the cells have the same passage number. The correlation see Answer for question 4.

  1. L161-165: PDL data for Tie2+ or Tie2- cells was not statistically significant, yet

statements of similar or inhibitory effect are made.

Answer: We agree the statements here are lacking strong statistical support. We assume the reason of not being significant is because of donor variance. The statement is based on the p-value with Kruskal-Wallis signed-rank test. The higher the p-value, the higher the possibility to reject the hypothesis of a difference between the two groups. The p-value between the Tie2+ PDL of spheroid group and the control/gelatin group is > 0.9999; consequently, the results are as similar. The p-value of Tie2- PDL of the spheroid group compared to the control group is 0.1547, of gelatin group, is 0.0649, so we can state an inhibitory effect. We added the respective p-value on L165-166.

  1. L185: What do the authors mean with “both”?

Answer: We apologize for this typo and deleted it.

  1. L208-210: Have the authors tried further rounds of spheroid generation? Will this

increase the % of Tie2 positive cells above 50% or even reach a pure Tie2 positive

cell population?

Answer: We haven’t tried further rounds of spheroid generation at present. We expect the further round spheroid regeneration will pure a Tie2+ population; this point will be the next step.

  1. L215-217: This statement should be supported by the demonstration of co-expression of Tie2 and a “stemness” and/or cell proliferation marker in the same cell.

Answer: Thanks for pointing that out. We deleted this overstatement, and we added your comment.

  1. Section 2.2: What was the percentage of Tie2-positive cells at day0?

Answer: In section 2.2, we compared the cells of the same passage cultured on different surfaces, so we did not show this data. The Tie2% on day0 is 5 ± 9 (means ± SD), and it was used to calculate the PDL of figure 3c of section.

  1. Section 2.5 is very confusing. Where do the “cells before seeding” come from? Are they just a thawed vial of frozen cells? The strategy and rationale behind this experiment are not clear from the way this paragraph is currently written. Why was day 10 monolayer culture designated as a control? Please rephrase and clarify. Maybe use Fig 8 here to clearly indicate what type of cells are used and compared for gene expression levels and explain why these choices were made. Clarify clearly what exactly day 0 cells are and why day10 monolayer cultured cells were chosen as control.

Answer: To improve clarity and avoid misunderstanding, we deleted this paragraph.

  1. L229: “and the gene expression of ACAN, COL1, COL2, TEK decreased significantly” during which time frame? Does this refer to the data in Fig 7?

Answer: To improve clarity and avoid misunderstanding, we deleted this paragraph.

  1. Section 3.3 seems quite speculative. L296/297: Please reword: “during culture, the

cells on the gelatin-coated surface were always reaching confluency in a faster way.”

L300-302: Please provide data for a cell proliferation marker or apoptosis data to

support this.

Answer: Thanks for point out. We also feel this section is too speculative and was lacking support. Therefore, we deleted this section.

  1. L309/310 Tie2 lineage analysis should be provided to support or exclude this

hypothesis.

Answer: We thank you for your comment. We would like to investigate the Tie2 lineage analysis in the future. We added this on L295.

  1. L312-314 This should be demonstrated by RNA in situ hybridization

Answer: We thank you for your comments, we added this point on L300,301.

  1. L318-320 The rationale behind this hypothesis should be elaborated further in general and in the context of the results presented.

Answer: Thank you for pointing this out. To improve clarity and avoid misunderstanding, we deleted this paragraph.

  1. What is the difference between NPC and NP phenotype cells?

Answer: NPCs are the cells, which are isolated from NP tissues. The NP phenotype cells are the cells that could produce ECM to maintain the NP tissue phenotype.

  1. L336/337 While the increase in Col2 expression is interesting, the Col2 expression

data in Fig7 was not statistically significant and the spread of data points and the large

SD makes this an overstatement.

Answer: Thanks for pointing this out. According to the reviewer’s comment, we deleted this overstatement.

  1. L451: Are these human or mouse genes?

Answer: Thanks for pointing this out. We added a modification on L426 to clearly reference to human genes.

  1. Gelatin should be sterile filtered, not autoclaved.

Answer: Thanks for point the lack of method. However, we strictly followed the protocol from “Redondo-Castro, E.; Cunningham, C.J.; Miller, J.; Cain, S.A.; Allan, S.M.; Pinteaux, E. Generation of Human Mesenchymal Stem Cell 3D Spheroids Using Low-binding Plates. Bio-Protocol 2018, 8, doi:10.21769/BioProtoc.2968.”, where the gelatin is autoclaved. We added this protocol to the references and cited it to clarify.

Round 2

Reviewer 2 Report

see attached file

This manuscript is a resubmission of an earlier submission. The following is a list of the peer review reports and author responses from that submission.

Round 1

Reviewer 1 Report

The article by Xinshuo Zhang et al. on Spheroid-like Cultures for Cell Expansion of Angiopoietin Receptor-1 (aka Tie2) positive cells from the human Intervertebral Disc touches on an important topic: Frequently used 2D culture is not representative of cells in vivo. The authors describe a spheroid-type culture method for the enrichment of nucleus pulpous progenitor cells (NPPC). While work from the Gantenbein group is typically of high quality, this paper is currently not acceptable for publication.

Some of the major points that need to be addressed:

The authors only use Tie2 as a marker to identify NCCP, which appears an oversimplification and hence they should refer to Tie2 positive cells instead of NPPC. The paper is extremely difficult to read due to the lack of the proper use of the English language and the authors might benefit from a native speaker proof-reading their manuscript prior to the next submission. Example: L42,113,115,192, 213, 217… .

The results are described in a fairly confusing manner. Tables should be included to summarize the results and help the reader to compare the different experiments.

The images in figures 2c and 2d are of poor quality and would benefit from a nuclear counter stain to indicate multicellularity.

L247: “NPC spheroids expressed the same number of ECM components as the native NP”. This is quite an overstatement, as only 2 components were assayed for.

While there are interesting points in the discussion, it is close to being incomprehensible due to the lack of proper English and it is not really reflecting much the findings of the actual study performed, example: Why is gelatine decreasing the number of Tie2 positive cells? In how far and why did the spheroid assay inhibit the proliferation of Tie2 negative cells?

The title of 3.3 (L272) should be rewritten and the spelling and wording throughout the entire paragraph should be revisited. Section 3.4: Please define characteristics of organoids. Revisit spelling and wording for L337-347. Section 3.7: Rewrite L371/372.

There is an inconsistency in the statistical analysis used for similar types of results without further reasoning, example Fig 6c versus Fig5b or 1a. Analysing some of the data with a z-proportion test might be better suited to compare subpopulations.

Given that this is a cell culture based experiment, the overall number of replicates seems very low, especially given some fairly wide spread results with a large SD, example: Figs 1b, 5b, 6b. More experiments should be carried out to strengthen the results.

The quality of the fluorescent images in figures 4 and 7 is quite poor. Are there any no probe controls for Figure 7?

The Material and methods section is inconsistent and incomplete. Examples: (1) Supplier name and product # should be provided for all material used including culture flasks and plates, FCS, Tie2 antibody … . (2) Procedure description and concentration of DAPI for live cell staining should be provided. (3) Information about the human discs from which the cells were derived should be provided: Type (lumbar versus cervical), stage of degeneration, age and sex of the patient. (6) On Average: How many cells per IVD could be derived? (7) A clear description of how the NP was defined in the human tissue samples and how the authors ensured that no TZ or AF cells were included in the studied cell populations is necessary. (8) A detailed protocol for the cell-freezing and thawing procedure should be provided. NOTE: Use of the word defrosting in this context is incorrect. (9) The source of the HUVEC cells is listed as gift in the acknowledgement and Thermo Fisher in the MM section, please clarify. (10) A detailed protocol of the immunostaining procedure should be provided. (11) Why is part 4.6 italicized? (12) Is this just one of the many typos or what is meant by “more than or equal to ten cells were counted” in L442? Do the authors mean spheres? (13) What was the magnification for undertaking the manual cell count? Did the same person conduct all counts to prevent inter person variability? (14) The gelatine bloom and the way the 0.1% gelatine solution was prepared should be provided, especially given the unexpected results. (15) How was a total cell count established?

Abbreviations should be explained the first time they are used in the text, example L27, L181, L406.

Also, the authors ignore significant work by others in the field, especially pioneers in the IVD field as well as work on the heterogeneous nature of IVD cells, including gene expression and stem cell markers of IVD cells in vivo and in 2D culture.

Author Response

Author replies to the queries of reviewers

Reviewer: 1

  1. The authors only use Tie2 as a marker to identify NCCP, which appears an oversimplification and hence they should refer to Tie2 positive cells instead of NPPC. The paper is extremely difficult to read due to the lack of the proper use of the English language and the authors might benefit from a native speaker proof-reading their manuscript prior to the next submission. Example: L42,113,115,192, 213, 217… .

Answer: We agree with the reviewer that we only focused on a single marker, which was CD202b, aka. Angiopoetin-receptor-1 aka Tie2. Thus, we replaced all mentionings of NPPC with Tie2+ NPCs where applicable. Concerning the English language, prior to final submission of the current manuscript, we submitted it to a professional English proof-reading agency.

  1. The results are described in a fairly confusing manner. Tables should be included to summarize the results and help the reader to compare the different experiments.

Answer: We went through the entire results section and re-structured the results. We thank the reviewer for pointing this out. We added two additional tables to summarize the main results of the manuscript.

  1. The images in figures 2c and 2d are of poor quality and would benefit from a nuclear counter stain to indicate multicellularity.

Answer: We thank the reviewer for pointing out this weakness of our methodology. Unfortunately, as the reason of COVID-19 sanitary crisis, our institute was emergency locked-down and the experiments related to this manuscript were stopped. Unfortunatly, we could not repeat it within this short period of time.

  1. L247: “NPC spheroids expressed the same number of ECM components as the native NP”. This is quite an overstatement, as only 2 components were assayed for.

Answer: We apologize for this unclear statement. We specified this by “two of the main native NP tissue ECM contents, i.e. aggrecan and collagen type II, were observed on NPC spheroids.” We also added a discussion of other components in lines L353-355.

  1. While there are interesting points in the discussion, it is close to being incomprehensible due to the lack of proper English and it is not really reflecting much the findings of the actual study performed, example: Why is gelatine decreasing the number of Tie2 positive cells? In how far and why did the spheroid assay inhibit the proliferation of Tie2 negative cells?

Answer: We thank the reviewer for pointing this out. We agree these two questions is worth discussing and we added it on the Lines L403-407.

  1. The title of 3 (L272) should be rewritten and the spelling and wording throughout the entire paragraph should be revisited. Section 3.4: Please define characteristics of organoids. Revisit spelling and wording for L337-347. Section 3.7: Rewrite L371/372.

Answer: We rewrote the section 3.3 (L272) in the section 3.2 on the Line L291-310. We define the characteristics of organoids in the introduction on the Line L92-95. In section 3.4, we would like to discuss the possibility of defining NPC spheroids as organoids. We rechecked and corrected L337-347 on the Line L357-367. We rewrote L371/372 on the lines L392/393.

  1. There is an inconsistency in the statistical analysis used for similar types of results without further reasoning, example Fig 6c versus Fig5b or 1a. Analysing some of the data with a z-proportion test might be better suited to compare subpopulations.

Answer: For the purpose of keeping consistency in the statistical analysis we sticked to a single way to perform the testing. We analyzed the data with multiple t-test (unpaired) which will non-integer performs many unpaired t tests at once -- one per row. Because Graphpad could not performed z-test with non-integer data and our experiment is only N=3-5 whereas the z-test usually need sample more than 30, we did not use z-test.

  1. Given that this is a cell culture based experiment, the overall number of replicates seems very low, especially given some fairly wide spread results with a large SD, example: Figs 1b, 5b, 6b. More experiments should be carried out to strengthen the results.

Answer: We are certainly aware of the relative low sample/donor repeats. Unfortunately, due to a complete lab lock-down due to COVID-19 pandemia we could not conduct additional replicates and all experiments related to this manuscript had to be stopped. We regret not being able to address this comment and that we could not repeat the experiment within the short time for revision.

  1. The quality of the fluorescent images in figures 4 and 7 is quite poor. Are there any no probe controls for Figure 7?

Answer: We apologize for the poor quality of the image in Figure 4 and Figure 7. We performed new image acquisition based on frozen cryosections to improve the quality of Figure 7. We also added the method accordingly in the Material and Methods section. Unfortunately, as the reason of COVID-19, the experiment related to this manuscript had to be stopped. Thus, we could not repeat the Figure 4, but we adjusted the brightness of the pictures, all consistently with the exact same filter. As for figure 7 we managed to re-stain frozen spheroids with two antibodies, ACAN and collagen type 2 and a DAPI counterstain. We have added these additional data in Figure 7.

The Material and methods section is inconsistent and incomplete. Examples:

  1. Supplier name and product # should be provided for all material used including culture flasks and plates, FCS, Tie2 antibody … .

Answer: We added this information on the new version of the manuscript.

  1. Procedure description and concentration of DAPI for live cell staining should be provided.

Answer: We apologize for the unclear description. The DAPI was used to stain dead cells for flow cytometry. We had clarified the Result part mentioning that spheroids were fixed before staining on line L178 and added the concentration of DAPI in the revised material and methods.

  1. Information about the human discs from which the cells were derived should be provided: Type (lumbar versus cervical), stage of degeneration, age and sex of the patient.

Answer: We added a table of the donors’ data to the current revision as Table 3.

  1. On Average: How many cells per IVD could be derived?

Answer: We added the data of tissues weight and the number of cells isolated for each donor. We got fragments from surgery of patient undergoing trauma. The cell density of NP is not even, cells are present with a higher density in the periphery and are present in a lower density in the center of the NP. Where the IVD fragments were removed by surgery, the individual difference of cell density per IVD were in fact unconfirmed due to the unprecise localization.

  1. A clear description of how the NP was defined in the human tissue samples and how the authors ensured that no cells from the transition zone (TZ) or annulus fibrosus (AF) cells were included in the studied cell populations is necessary.

Answer: The tissue was pre-separated by experienced spine surgeons, i.e. by C.A. and S.H.. Furthermore, the tissue was pre-selected under macroscopic digestion before tissue preparation. The cells were isolated with ISO 9001 certified SOPs, which were developed by experts at the AO Research Insitute in Davos and were further fine-tuned by the group head and lab technicians at UBern. The general procedure of cell isolation followed the article described by Lee et al. (2015). J Orthop Res 33(12):1743-1755 https://doi.org//10.1002/jor.22942. Of course, one can never exclude some cells form the TZ to be in the digestion. We added this information on the Line L432-434.

  1. A detailed protocol for the cell-freezing and thawing procedure should be provided. NOTE: Use of the word defrosting in this context is incorrect.

Answer: We added the methodology of the cell-freezing and thawing to section 4.3 and changed the wording of defrosting to thawing.

  1. The source of the HUVEC cells is listed as gift in the acknowledgement and Thermo Fisher in the MM section, please clarify.

Answer: We apologize for the unclear description. The source of the HUVEC was kindly provided by a colleague of the AO Research Institute. We specified this in the methods section to be consistent with the acknowledgements section.

  1. A detailed protocol of the immunostaining procedure should be provided.

Answer: We apologize for not providing enough details of immunostaining. We have now  added these details to the Material and Methods part.

  1. Why is part 4.6 italicized?

Answer. This was a formatting mistake on our side. We changed it.

  1. Is this just one of the many typos or what is meant by “more than or equal to ten cells were counted” in L442? Do the authors mean spheres?

Answer:  We apologize for the unclear description. “more than or equal to ten cells were counted” referred specifically to the CFU-assay. In this assay, only cells number equal or superior to 10 were counted as a single colony. We clarified it on lines L509-510 and the following.

  1. What was the magnification for undertaking the manual cell count? Did the same person conduct all counts to prevent inter person variability?

Answer: The cell count was performed under a magnification of 10X. The cell count was taken by one experimentor (the first author of the manuscript, X. Z.). Unfortunately, we have not considered independent counters. We will consider this input for future study designs. We have added these details on lines L493/494.

  1. The gelatine bloom and the way the 0.1% gelatine solution was prepared should be provided, especially given the unexpected results.

Answer: We have added the details in section 4.2.

  1. How was a total cell count established?

Answer: As mentioned in point 20 the cell count was taken by one person using a improved Neubauer counting chamber. We apologize for providing few details about this. We gave the details in section 4.2

  1. Abbreviations should be explained the first time they are used in the text, example L27, L181, L406.

Answer: Thank you for pointing out. We apologize for the missing abbreviation explanation. We explained the abbreviations you pointed out on the line L28,190,456

  1. Also, the authors ignore significant work by others in the field, especially pioneers in the IVD field as well as work on the heterogeneous nature of IVD cells, including gene expression and stem cell markers of IVD cells in vivo and in 2D culture.

Answer: We thank the reviewer for this statement and critics. We have added additional references to the introduction (L67-69) and have also added a new section to the discussion to appreciate previous work on 3D cell culture and IVD markers, and also cited classical works on IVD regeneration(L324-326).

Reviewer 2 Report

In their manuscript „Spheroid-like Cultures for Cell Expansion of Angiopoietin Receptor-1 (aka Tie2) positive cells from human intervertebral disc” Zhang et al. compare different culturing methods of nucleus pulposus cells (NPC) and nucleus pulposus progenitor cells (NPPC).

The manuscript is well structured but requires a careful linguistic revision since it is hard to read, some sections are difficult to understand and the quintessence remains somewhat unclear. Also, numbering of paragraphs should be carefully revised. However, I strongly encourage the authors to address the following points.

  1. Discussion section, page 10, lines 295-296: Authors state that the amount of Tie+ cells in spheroid culture was twice as high as in monolayer culture. The statement seemingly refers to Figure 1a, but in the graph Tie+ cells in spheroid culture were 3 to 5-fold the amount of Tie+ cells in monolayer culture. The same applies to a statement on page 10, lines 298-299 (“spheroids on suspension-culture of ultra-low attachment surfaces formed about four times more colonies that were identified as CFU-s than from monolayer”), whereas in the corresponding Figure 2A the difference is only 2 to 3-fold. Please clarify and in favor of the reader explicitly refer to the corresponding Figures in the Discussion section.
  2. Figure 1 shows that cultivation of NPCs on ultra-low surface results in a larger fraction of Tie2+ positive NPPC (26%) while it seems to inhibit growth of Tie- NPCs. Given this growth inhibition, I would have expected a larger fraction of the cells to be Tie2+. Are there any data available regarding the further characterisation of the remaining Tie- cells?
  3. As far as I understood, the aim was to improve NPC cultivation as such that NPPCs are enriched. To what extent are antibody-based enrichment methods (e.g. magnetic beads) an option to isolate or purify Tie2+ cells?
  4. In Figure 2 the formation of different colonies (spheroid, fibroblast-like, mixed) was assessed in relation to previous culture methods. It is nicely shown that cultivation on ultra-low attachment surfaces results in increased formation of spheroid colonies (CFU-s). However, given the low fraction of Tie+ cells in populations grown on standard plastics or gelatine-coated plastics, one would expect a tendency of these cells to rather form CFU-f instead of CFU-s. However, according to Figure 2a, these populations seem not to form any colonies at all. Can you please discuss?
  5. Section 2.3 (CFU-s formation after de-frosting): it is not fully clear on which surface cells were cultivated before freezing and on what surface the cells were allowed to recover after de-frosting before subjected to CFU assay (defrost-monolayer). Also, on page7, line 199 it is stated that NPPC-yield from defrost-monolayer group is 6% less than from directly-defrosted group. Data shown in Figure 5b are the very reverse of this statement.
  6. On page 8, lines 224-226 it says: “cells from 1st-generation-spheroids formed 89 ± 49 (mean±SD) colonies identify as CFU-s, and from 2nd -generation-spheroids formed 30 ± 28 colonies identity as CFU-s”. Again, data presented in Figure 6d show the exact opposite. Also, the ability to form CFU-f and CFU-s/f (NOT CFU-s and CFU-f/s) is similar between 1st and 2nd generation spheroids. These discrepancies between figures and text are irritating and are not appropriate for a proper scientific manuscript.
  7. Section 2.6 (Extracellular matrix of NPCs spheroids). Fluorescent staining of ECM components aggrecan and collagen type 2 was performed on NPC spheroids. Unfortunately, the pictures in the manuscript are of poor quality. Thus, evaluation of staining and component distribution is quite a challenge. Figure 7 suggests that staining for each component were performed individually. If this is the case, I would recommend to perform a double staining, otherwise merge pictures are more appropriate to show. Further, while on page 8, line 247-248 it is stated that “NPC spheroids expressed the same number of ECM components as the native NP tissue”, while in the discussion section authors claim that “The proportion of collagen type 2 and aggrecan is essential for the NP tissues; however, this was not assessed in this study“ (page11, lines 331-332). Please clarify this discrepancy.
  8. In the discussion section authors state that as a result of their study, the suspension environment is not indispensable for NPPC expansion. This is somewhat ambiguous. From the data presented in this manuscript I understood that the suspension environment (cultivation ultra-low attachment surface) enables an enrichment of Tie2+ NPPCs and results in increased ability to form spheroid colonies (CFU-s). I strongly recommend to re-write this section for more clarity and non-ambiguous conclusions.
  9. I strongly encourage the authors to provide any additional data (e.g. enzymatic digestion described in paragraph 3.5) as well as raw data (e.g. FACS populations and gating) in supplements.

Author Response

Reviewer 2

  1. The manuscript is well structured but requires a careful linguistic revision since it is hard to read, some sections are difficult to understand and the quintessence remains somewhat unclear. Also, numbering of paragraphs should be carefully revised. However, I strongly encourage the authors to address the following points.

Answer: We thank the reviewer for this comment. We have revised the majority of the main text and the conclusions. We went carefully through the text and have rewritten major parts of the results and hopefully, made the conclusions much clearer. The numbering of paragraphs was corrected following the comment of the Reviewer. The text was also proof-read by a native English speaker.

  1. Discussion section, page 10, lines 295-296: Authors state that the amount of Tie+ cells in spheroid culture was twice as high as in monolayer culture. The statement seemingly refers to Figure 1a, but in the graph Tie+ cells in spheroid culture were 3 to 5-fold the amount of Tie+ cells in monolayer culture. The same applies to a statement on page 10, lines 298-299 (“spheroids on suspension-culture of ultra-low attachment surfaces formed about four times more colonies that were identified as CFU-s than from monolayer”), whereas in the corresponding Figure 2A the difference is only 2 to 3-fold. Please clarify and in favor of the reader explicitly refer to the corresponding Figures in the Discussion section.

Answer: We sincerely apologize for these mistakes. The fold times in discussion section should like your mentioned. We have corrected them on Line L314,316

  1. Figure 1 shows that cultivation of NPCs on ultra-low surface results in a larger fraction of Tie2+ positive NPPC (26%) while it seems to inhibit growth of Tie2- NPCs. Given this growth inhibition, I would have expected a larger fraction of the cells to be Tie2+. Are there any data available regarding the further characterisation of the remaining Tie- cells?

Answer: We thank the reviewer to point this out, we added a discussion about this problem on the Line L397-402. Unfortunately, we did not have further characterization of Tie2- cells except negative to Tie2 marker. We added how to define Tie2- cells of figure 4 in method section on the Line L526,527

  1. As far as I understood, the aim was to improve NPC cultivation as such that NPPCs are enriched. To what extent are antibody-based enrichment methods (e.g. magnetic beads) an option to isolate or purify Tie2+ cells?

Answer: Previous studies by our group (Frauchiger, et al. Fluorescence-Activated Cell Sorting Is More Potent to Fish Intervertebral Disk Progenitor Cells Than Magnetic and Beads-Based Methods. Tissue Engineering Part C-Methods, 2019), which focused specifically how Tie2+ cells can be best isolated clearly demonstrated that FACS method is superior over Magnetic bead-based or mesh-based methods to isolate these cells at least in a bovine animal model (Frauchiger et al. (2019) Tissue Eng Part C Methods 25(10):571-580. However, we have added this option as a discussion point in the lines L59-62.

  1. In Figure 2 the formation of different colonies (spheroid, fibroblast-like, mixed) was assessed in relation to previous culture methods. It is nicely shown that cultivation on ultra-low attachment surfaces results in increased formation of spheroid colonies (CFU-s). However, given the low fraction of Tie2+ cells in populations grown on standard plastics or gelatine-coated plastics, one would expect a tendency of these cells to rather form CFU-f instead of CFU-s. However, according to Figure 2a, these populations seem not to form any colonies at all. Can you please discuss?

Answer:  We added a discussion on the Line L397-402.

  1. Section 2.3 (CFU-s formation after de-frosting): it is not fully clear on which surface cells were cultivated before freezing and on what surface the cells were allowed to recover after de-frosting before subjected to CFU assay (defrost-monolayer). Also, on page7, line 199 it is stated that NPPC-yield from defrost-monolayer group is 6% less than from directly-defrosted group. Data shown in Figure 5b are the very reverse of this statement.

Answer: We added the detail about surface on the Line L198-200. We have corrected the mistake on Line L211.

  1. On page 8, lines 224-226 it says: “cells from 1st-generation-spheroids formed 89 ± 49 (mean±SD) colonies identify as CFU-s, and from 2nd -generation-spheroids formed 30 ± 28 colonies identity as CFU-s”. Again, data presented in Figure 6d show the exact opposite. Also, the ability to form CFU-f and CFU-s/f (NOT CFU-s and CFU-f/s) is similar between 1stand 2nd generation spheroids. These discrepancies between figures and text are irritating and are not appropriate for a proper scientific manuscript.

Answer: We have corrected these mistakes on Line L235 and L236.

  1. Section 2.6 (Extracellular matrix of NPCs spheroids). Fluorescent staining of ECM components aggrecan and collagen type 2 was performed on NPC spheroids. Unfortunately, the pictures in the manuscript are of poor quality. Thus, evaluation of staining and component distribution is quite a challenge. Figure 7 suggests that staining for each component were performed individually. If this is the case, I would recommend to perform a double staining, otherwise merge pictures are more appropriate to show. Further, while on page 8, line 247-248 it is stated that “NPC spheroids expressed the same number of ECM components as the native NP tissue”, while in the discussion section authors claim that “The proportion of collagen type 2 and aggrecan is essential for the NP tissues; however, this was not assessed in this study“ (page11, lines 331-332). Please clarify this discrepancy.

Answer: We acquired new fluorescent images and have improved the image quality of figures 4 and 7. We have corrected the imprecise description on Line L258,259, and 351-352.

  1. In the discussion section authors state that as a result of their study, the suspension environment is not indispensable for NPPC expansion. This is somewhat ambiguous. From the data presented in this manuscript I understood that the suspension environment (cultivation ultra-low attachment surface) enables an enrichment of Tie2+ NPPCs and results in increased ability to form spheroid colonies (CFU-s). I strongly recommend to re-write this section for more clarity and non-ambiguous conclusions.

Answer:  We clarified the discussion  on L287.

  1. I strongly encourage the authors to provide any additional data (e.g. enzymatic digestion described in paragraph 3.5) as well as raw data (e.g. FACS populations and gating) in supplements.

Answer: We now have included the gating strategy as supplementary online Figure on the FACS sorting procedure of the human NPC (S2). Moreover, the enzymatic digestion (S1) protocol has been added too.

Round 2

Reviewer 1 Report

Ijms-820636 peer review vs2

Some examples of typos/grammatical errors:

L199: The frozen cells were in passage one after cultured in a monolayer on “classic” plastics.

L200: One passage recovery after thawing improved the ability NPCs in the next passage to form CFU-s colonies from NPC spheroids on ultra-low attachment surfaces.

L202: Then the NPCs on passage two assessed the CFU assay and the Tie2+ NPC yield

L206: Next, the NPCs on passage three assessed the CFU assay and the Tie2+ NPC yield.

L217: 0.1% gelation-coated T75 flask

L224: feasibility of NPCs 'to pass from suspension culture on ultra-low attachment surface

Table1:

  • Spheroid formation inhibition Tie2-NPC proliferation but, did not improve the proliferation compared with monolayer culture (control and gelatin groups).
  • After thawing, it is better to rest the cells one passage (monolayer culture) and used the resuspended cells to form spheroids.
  • The NPCs from spheroid shown the higher ability of CFU-s formation compared
  • Tie2% of NPC dose not influence by using the cells thawing directly
  • The NPCs from 2nd-generation-spheroids shown higher ability of CFU-s formation

Table2: The collagen type 2 and aggrecan is are the main contents of NP tissue.

L282: including the Tie2+ Tie2+ NPCs

L291: In 1992, Reynolds and Weiss in 1992 described

L301: Spheroid formation also used to isolate stem cells.

L326: both new primary cells (data do not show)

L335: The papain (10 I.U./ml) could not help digestion a total of 5 minutes.

L336: The upper limit to passages of spheroid formation in this study were up to two passages in suspension culture

L352: Before define NPC spheroids as organoids,

L374: In this study, only bFGF was used to keep the phenotype of Tie2+ NPC.

L412: which part of the Tie2+ NPCs and Tie2- NPCs is responsible for the expression of the matrix should be clarified.

L450: Before coated,

L452: If gelatin-coated flasks were not used directly, it was stored at

L523: pictures of Tie2 stain were taken

L529/530: with centrifuge at 500 g of 5 min / and centrifuge at 500 g of 5 min

L532: the freeze spheroids were stored at -150°C

L534: The Frozen section

Unclear or incomplete sentence/statement:

L232/233: The Tie2+ NPC yield increased to 26% by passaging and reforming spheroids

L239/241 Passaging NPCs with spheroids formation assay increased the yield of Tie2+NPCs two times, and a relatively purified NPPCs population compared to the NPCs assess spheroid formation assay one time.

Table1: NPC spheroids with Tie2 staining. Note: HUVEC showed Tie2 staining also – what is the take home message?

Table2: I don’t think this table is very helpful. This information can be added to the figure legends.

L286-288: The reason for the lower Tie2+ NPC yield from the monolayer culture on both classic and gelatin-coated surfaces compared with spheroid culture is that there are different proliferation levels between Tie2+ NPCs and Tie2- NPCs.

L311/312: the amount of  Tie2+ NPCs in spheroid culture was represented about three to five fold of it in monolayer culture.

L319: The Tie2+ NPCs proliferation rate did not improve under equal time

L323/324: Recently, smart biomaterials were developed that not only facilitate progenitor status of IVD cells in general by incorporating laminin [38].

L354: the culture situation, such as the limitation of the diameter of the spheroid due to the metabolism supply limitation, should be assessed in the future.

L355: the fission of metabolism

L358: The character of avascular gives us another supplemental way to identify NPC spheroids as NPC organoids.

L364:  why the clonal formed in degeneration NP could not stop the degeneration will be answered

L404-406: Thus, we assume that when there are enough Tie2- NPCs, a mechanism like a contact inhibition will inhibit the proliferation of Tie2+ NPC. This assumed mechanism will avoid the accumulation of non-NP phenotype cells differentiated from Tie2+ NPC. Please elaborate more on the selectivity of this assumed mechanism

L413-415: There are three hypotheses: i) Tie2+ NPCs also express matrix, ii) only the young Tie2 negative NPCs differentiated from Tie2+ NPCs can help expression matrix, and iii) Tie2 negative NPCs' matrix expression is dependent on Tie2+ NPCs' regulation.How do the results of this paper lead to these hypotheses and which one is most likely supported by the results of this paper?

L421: But the upper limit of the purification ability still needs further research.

L466/ L517: Trypsin-EDTA Solution – concentration?

Sample number too low, SD very large – remove or repeat

L210 /211: However, the Tie2+ NPC yield in the thawed-monolayer groups had 6% more  of Tie2+ NPCs compared to the group that was directly thawed (21 ± 15) (mean ± SD) (Figure 5b).

L221/FIGURE 5B: N=3 (One donor lost in Thaw-directly group of spheroids).

Sample number too low

L525: Three images of different sites of the slide were collected, and Tie2+ stained cells were counted manually (by XZ). The cells with Tie2-PE signaling around the nucleus (DAPI) were counted as Tie2+ NPC, the other cells were identified as Tie2-NPC.  - Low sample number, for a similar analysis please see Li et al (2018) DOI: 10.1111/joa.12904 .

Confirm stemness through the expression of at least one pluripotency marker like Oct4 or Sox2 L241-243: This further passaging protected the stemness of Tie2+ NPCs, which was confirmed by an increase of the self-renewal ability cells of the whole NPC population.

Quality of Figure 4a is poor. We all understand the difficulties COVID19 brought upon all of our research, however, this does not justify for substandard work to be accepted for publication. There are several occasions in this manuscript where data is listed as “not shown”. Why showing something here, that really does not increase confidence in the results derived from it. I do not see 4b supported by 4a, especially when assuming the image shown is best of each of the three images used for counting.

?? - Discussion: L285/286: As a result, the suspension environment is not necessary for the Tie2+ NPCs' expansion. Conclusion: L555: Expansion in spheroid-formation-culture could potentially increase the number of Tie2+ cells after prolonged cell culture of Tie2+NPCs.

Reviewer 2 Report

In the second version of the manuscript, authors addressed some of the remarks made by the reviewers and improved the manuscript. Still, there are a some major points that should be addressed.

  • Tie2 staining and quantification in FACS. When comparing the cell population in supplementary figure S2e and S2f (isotype control vs Tie2 stain) it seems that whole population is slightly shifted to the right. This is often to be observed when many cells in a population express low amounts of the antigen. Therefore, it seems that far more cells are Tie2 positive than the gated 6.7%. I suggest to analyse the mean/median of PE (isotype control vs stain) and to statistical analyse the difference in biological replicates.

  • Data shown are replicates of n=3, except for spheroid; thaw-directly group (due to donor loss). Given the large variation of the data in this group, I strongly recommend verification with another biologic replicate.

  • The statement in Discussion section L285 (“suspension environment is not necessary for the Tie2+ NPCs expansion”) contradicts the shown data (and the conclusion drawn in L 553). I do not understand these discrepancies.

  • I do understand that the lockdown due to the Corona crisis caused some difficulties in the lab work. Still, conclusions drawn in the manuscript should be proven by any available data. Therefore, I am sorry if I haven’t expressed my self clearly: I strongly encourage the authors to show all their collected data (L182 and 277) and to repeat experiments/ increase replicates where necessary.

  • Please check again carefully grammar and style of the manuscript. Some examples are given below:
    • Consistent labelling of figures and axes (e.g. HUVEC vs huvec)
    • Sub paragraph 2.6 in results section header seems wrong.
    • Incomplete /Unclear sentences: L147, 158-160, 198, 282, 364, 555, etc